# Interleukin6 prediction of mortality in critically ill COVID19 patients: A prospective observational cohort study

**Amira Jamoussi** [1,2] *, **Lynda Messaoud** [1,2], **Fatma Jarraya** [1,2], **Emna Rachdi** [1,2], **Nacef Ben Mrad** [1,2], **Sadok Yaalaoui** [3], **Mohamed Besbes** [1,2], **Samia Ayed** [1,2], **Jalila Ben Khelil** [1,2]

**1** Faculty of Medicine, Medical Intensive Care Unit, Abderrahmen Mami Hospital, University of Tunis El Manar, Ariana, Tunisia, **2** Research Unit for Respiratory Failure and Mechanical Ventilation UR22SP01, Ministry of Higher Education and Scientific Research, Tunis, Tunisia, **3** Faculty of Medicine, Immunology Laboratory, Abderrahmen Mami Hospital, University of Tunis El Manar, Ariana, Tunisia

* amira.jamoussi@fmt.utm.tn

**Data Availability Statement:** All relevant data are within the manuscript and its Supporting information files.

## Abstract

### Objective

The aim of this study is to explore the role of IL6 in predicting outcome in critically ill COVID-19 patients. Design Prospective observational cohort study. Setting 20-bed respiratory medical intensive care unit of *Abderrahmen Mami Teaching Hospital* between September and December 2020.

### Methods

We included all critically ill patients diagnosed with COVID-19 managed in ICU. IL6 was measured during the first 24 hours of hospitalization.

### Results

71 patients were included with mean age of 64 ± 12 years, gender ratio of 22. Most patients had comorbidities, including hypertension (n = 32, 45%), obesity (n = 32, 45%) and diabetes (n = 29, 41%). Dexamethasone 6 mg twice a day was initiated as treatment for all patients. Thirty patients (42%) needed high flow oxygenation; 59 (83%) underwent non-invasive ventilation for a median duration 2 [1–5] days. Invasive mechanical ventilation was required in 44 (62%) patients with a median initiation delay of 1 [0–4] days. Median ICU length of stay was 11 [7–17] days and overall mortality was 61%. During the first 24 hours, median IL6 was 34.4 [12.5–106] pg/ml. Multivariate analysis shows that IL-6 $\geq$ 20 pg/ml, CPK < 107 UI/L, AST < 30 UI/L and invasive ventilation requirement are independent risk factors for mortality.

### Conclusions

IL-6 is a strong mortality predictor among critically ill COVID19 patients. Since IL-6 antagonist agents are costly, this finding may help physicians to consider patients who should benefit from that treatment.

**Funding:** This study was funded by the Tunisian Ministry of Higher Education and Scientific research (Project: TriOMICS-CoV: PRFCOV19-GP1).

**Competing interests:** The authors have declared that no competing interests exist.

## Introduction

Coronavirus disease 2019 (COVID-19) causes an inflammatory response and the degree of inflammatory cytokine storm is linked to COVID-19 related severity [1]. Cytokine storm is a maladaptive cytokine release in response to infection and other stimuli with a complex pathogenesis [2]. Studies describing the immunological profile of critically ill COVID-19 patients suggest hyperactivation of the humoral immune pathway Interleukin6 (IL6). Specifically, IL6 was highlighted to predict occurrence of respiratory failure, shock and multiorgan dysfunction [3]. It is now recognized that serum levels of IL-6 are significantly elevated in severe COVID-19 disease. Indeed, patients with complicated forms of COVID-19 had nearly threefold higher serum IL-6 levels than those with noncomplicated forms of the disease [3].

In the ICU, we need to rely on reliable outcome indicators as points of assessment. In critically ill COVID-19 patients, the ability of biomarkers to predict poor outcomes such as death is still under review.

The aim of the present study is to explore whether IL6 levels can predict outcomes. Secondarily, we aimed to search for correlations between initial IL6 levels and others clinico-biological parameters.

## Methods

### Study design and patients' selection

A single centre prospective cohort study was conducted in a dedicated COVID-19 ICU of *Abderrahmen Mami Teaching Hospital* between September and December 2020. Participants were recruited for this study from the inpatient ICU population with full verbal consent acquired from each patient prior to inclusion in our study.

Inclusion criteria included all critically ill patients diagnosed with COVID-19 by RT-PCR and managed in the ICU between September and December 2020.

**Study protocol.**   For all patients admitted to the ICU, IL-6 levels were measured during the first 24 hours. Quantitative determination of serum IL-6 levels was performed with a solid phase, enzyme labelled, chemiluminescent immunometric assay on an *Immulite 1000* analyser (Siemens™). Normal range of serum IL-6 quantification was under 2 pg/mL.

**Clinical data.**   We collected information on demographic characteristics, underlying diseases and chest CT imaging. Severity of illness was evaluated according to the Simplified Acute Physiology Score II (SAPS II) and the Acute Physiology and Chronic Health Evaluation (APACHE) II during the first 24 hours in the ICU. Management features such as steroids, the need for invasive mechanical ventilation and/or non-invasive mechanical ventilation, were also recorded. Laboratory results including arterial blood gases (ABG), lactates, blood cell counts, Zinc, C-reactive protein (CRP), fibrinogen, Ddimer, urea, creatinine, creatine-phospho-kinase (CPK), lactico-deshydrogenase (LDH), Aspartate aminotransferase (AST), Alanine aminotransferase (ALT), N-terminal pro-B type natriuretic peptide (NT pro BNP) and Troponin were reported for each patient. Lastly, outcome data such as length of stay and ICU mortality were recorded.

### Objectives

Primary outcome was to explore the role of IL6 in the prediction of COVID-19 related ICU mortality and invasive mechanical ventilation requirement. Secondary outcomes were to investigate IL-6's correlations with biological data also measured during the first 24 hours of hospitalization; and to search for all independent ICU mortality risk factors.

## Statistical analyses

SPSS 23.0. was used for statistical analyses. Descriptive statistics of the patients were calculated and reported in terms of absolute frequencies and percentages for the qualitative variables. Quantitative variables in our cohort according to the Kolmogorov-Smirnov test were predominantly non-parametric. Quantitative variables were expressed as either medians and IQR 25th and 75th percentiles or in terms of means ± standard deviation (SD). Analysis of the differences in clinical characteristics, biological data and management requirements between survivors and non-survivors was performed. The differences between independent groups regarding continuous variables were evaluated by Student's t-test. Nominal data were analysed by Pearson's Chi-square test or Fisher's Exact test, when appropriate. Medians of quantitative variables between groups were compared using the Mann-Whitney nonparametric test.

Optimal cut off values were also determined using receiver operating characteristic curve (ROC) analysis.

Variables which showed a significant p value in the univariate analysis were entered into the model. A logistic regression was performed to obtain an adjusted estimate of the odds ratios (ORs) and to identify which factors were independently associated with ICU mortality.

We tested the role of IL-6 as risk factor for negative outcome. The correlation between IL-6 and biological data was studied according to Spearman's coefficient and curves built if the correlation was significant. Data were considered to be statistically significant, if the p values were less than 0.05.

## Ethical considerations

All data used in the analysis were collected in the routine surveillance activities suggested during COVID-19 pandemic, so did not require informed consent. All data were fully anonymized before we accessed them. Study ethical approval was given by institutional review board (Abderrahmen Mami hospital's local committee) and waived the requirement for informed consent. Patients agreed to study results dissemination.

## Results

### Baseline characteristics

A total of seventy-one patients who were diagnosed with coronavirus and admitted to the ICU were included in this study, with a mean age 64 ± 12 years and gender ratio of 2,2. The reason for admission to the ICU was acute respiratory failure in all patients with a median $PaO_2/FiO_2$ ratio of 120 IQR [85–156] mmHg. Chest CT was performed in 45 (63%) patients showing ground glass opacities and/or consolidations in all cases. Pulmonary parenchyma damage extent was estimated to be >75% (n = 13), 50–75% (n = 10), 25–50% (n = 17) and < 25% (n = 5). Baseline characteristics and onset severity of all participants are further detailed in Table 1.

### Laboratory findings

At ICU admission, lymphopenia ($< 1.500 \ 10^3/mm^3$) was noticed in 54 cases (76%) of which 30 (42%) was very low ($<1.000 \ 10^3/mm^3$). Zinc serum deficiency ($<70$ μg/L) was recorded in 47 patients (%). Rhabdomyolysis (CPK > 1000 UI/l) was noticed in 5 patients (7%) and marked transaminases elevation in 9 patients (13%). Main laboratory findings of all participants are detailed in Table 2.

**IL-6 description.** Serum IL-6 concentration was high in all patients, median and IQR were 34.4 [12.5–106] pg/ml. IL-6 showed a significant correlation with age (r = 0.301, p = 0.011), CRP (r = 0.287, p = 0.021), zinc (r = -0.326, p = 0.011), and D-dimers (r = 0.281,

**Table 1. Baseline characteristics and onset severity.**

| | n = 71 |
|---|---|
| Age, mean ± SD, years | 64±12 |
| Gender, n (%) | |
|   Male | 49 (69) |
|   Female | 22 (31) |
| SAPSII, mean ± SD | 30 ± 10 |
| APACHEII, mean ± SD | 10 ± 6 |
| Comorbidities, n (%) | |
|   Hypertension | 32 (45) |
|   Diabetes | 29 (41) |
|   Obesity | 32 (45) |
| Symptoms delay, med [IQR] | 10 [7–13] |
| Acute hypoxemic respiratory failure, n (%) | 71 (100) |

p = 0.033) (Fig 1). We focused on IL-6 level association with outcome parameters (Table 3) which were significant for ICU mortality. Under ROC curve analysis (Fig 2), IL-6 $\geq$ 20 pg/ml was found to be significantly associated with ICU mortality (AUC = 0.805, p $< 10^{-3}$, sensitivity = 0.837 and specificity = 0.679).

## Management and outcome

Dexamethasone was the main treatment provided at a dose of 6 mg twice a day. All patients required respiratory assistance. Thirty patients (42%) needed high flow oxygenation; 59 (83%)

**Table 2. Laboratory findings during the first 24 hours of hospitalization.**

| | Median | IQR |
|---|---|---|
| White blood cells, $10^3/mm^3$ | 9.500 | 6.800–13.400 |
| Lymphocytes, $10^3/mm^3$ | 1.020 | 0.760–1.420 |
| Hemoglobin, g/dL | 13 | 11.6–14 |
| Platelets, $10^3/mm^3$ | 251 | 201–293 |
| CPK, UI/L | 114 | 56–233 |
| LDH, UI/L | 425 | 341–603 |
| AST, UI/L | 31.5 | 22–45.2 |
| ALT, UI/L | 27 | 20–42.5 |
| IL-6, pg/ml | 34.4 | 12.5–106 |
| Zinc, µg/L | 51.5 | 36.2–69 |
| CRP, mg/L | 119.5 | 54.8–187.7 |
| Troponin, ng/ml | 0.01 | 0–0.03 |
| NT-proBNP, pg/ml | 31.8 | 14–72 |
| Fibrinogen, g/L | 6.36 | 4.68–6.89 |
| D-Dimer, µg/L | 670 | 420–2950 |
| pH | 7.47 | 7.43–7.51 |
| $PaO_2/FiO_2$, mmHg | 120 | 85–156 |
| $PaCO_2$, mmHg | 36 | 32–42 |

CPK: creatine-phospho-kinase; LDH: lactico-deshydrogenase; AST: Aspartate aminotransferase; ALT: Alanine aminotransferase; IL-6: Interleukin-6; CRP: C-reactive protein; NT pro BNP: N-terminal pro-B type natriuretic peptide.

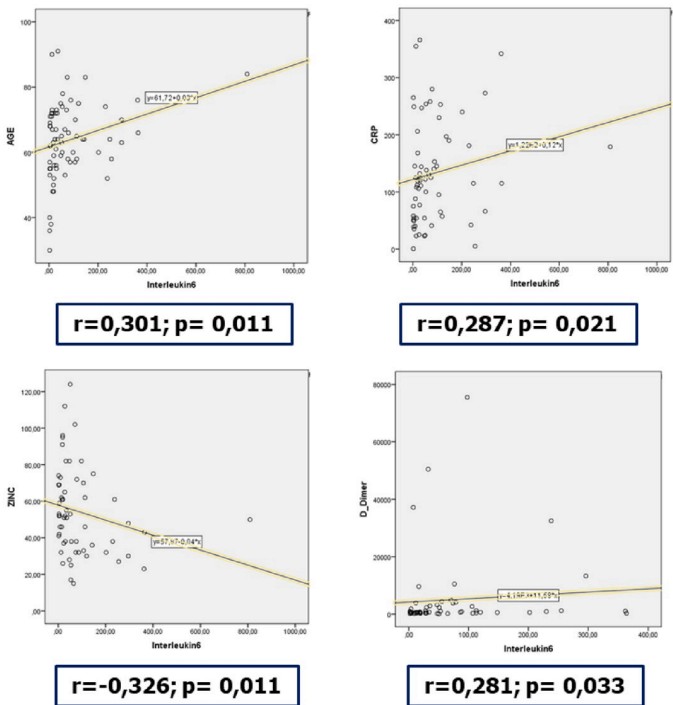

r=0,301; p= 0,011    r=0,287; p= 0,021

r=-0,326; p= 0,011    r=0,281; p= 0,033

**Fig 1. Serum IL-6 concentration correlation with age, C-reactive protein, zinc, and D-dimers (r = Spearman's coefficient).**

underwent non-invasive ventilation for a median duration 2 [1–5] days. Invasive mechanical ventilation was required in 44 (62%) patients with a median initiation delay of 1 [0–4] days. Median ICU length of stay was 11 [7–17] days and overall mortality was of 61%.

**Mortality analysis.** A comparison of the characteristics of survivors and non survivors is detailed in Table 4.

Multivariate logistic regression, detailed in Table 5, indicates that IL-6 $\geq$ 20 pg/ml, CPK < 107 UI/L, AST < 30 UI/L and invasive ventilation are independent risk factors for mortality.

## Discussion

In this prospective cohort study of critically ill COVID19 patients, we report 3 major findings. First, IL-6 is a strong mortality predictor, more accurately a cut-off concentration of IL-6 $\geq$ 20 pg/mL at initial assessment in ICU. Second, IL-6 has significant correlations with age, CRP, zinc and D-dimer. Third, independent risk factors for ICU mortality, in addition to IL-6, were invasive mechanical ventilation requirement, AST < 30 UI/L and CPK < 107 UI/L.

**Table 3. IL-6 level association with outcome.**

|  | IL-6 (pg/ml) | p |
|---|---|---|
| Invasive mechanical ventilation |  |  |
| Yes (n = 44) | 33.5 [15.7–111.5] | 0.214 |
| No (n = 27) | 35.6 [4.2–86.1] |  |
| Death |  |  |
| Yes (n = 43) | 70.8 [27.7–143] | $< 10^{-3}$ |
| No (n = 28) | 14.7 [3.4–33.3] |  |

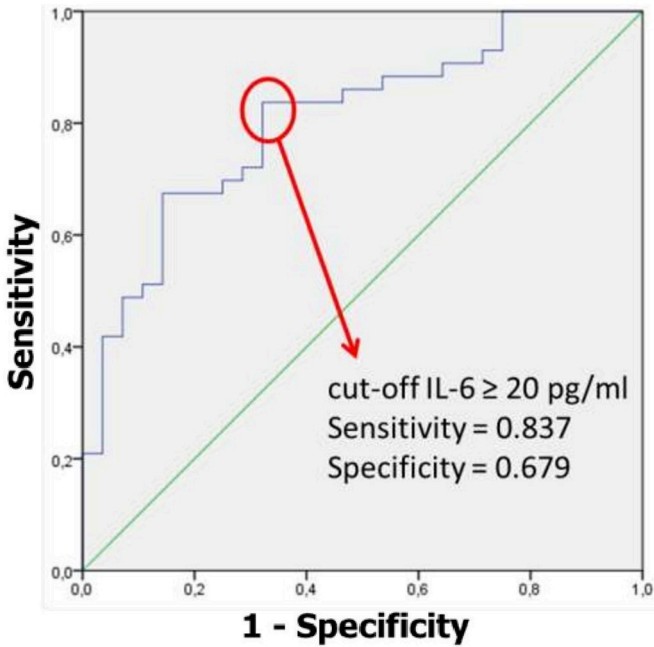

**Fig 2. Receiver operating characteristic (ROC) curve showing the predictive power of IL-6 for predicting ICU mortality among critically ill COVID-19 patients.**

Relevance of IL-6 in mortality prediction was already studied for several clinical conditions, other than COVID19. This biomarker has been shown to predict mortality in heart failure [4], haemodialysis patients [5], end-stage liver disease [6] or hospitalized patients with cancer [7].

There is a substantial body of evidence linking the IL6 concentration to the severity of disease and unfavourable outcome of Covid-19 [8–10]. In our study, the optimal cut-off value of IL-6 to predict mortality was 20 pg/mL. Grifonin E. et al [11] reported IL-6 of > 25 pg/ml to be a sufficient predictor for severe COVID19 and/or in hospital mortality. It should be noted that the authors had excluded patients requiring immediate ICU admission (indicating patients with a lesser degree of disease severity).

Targeting the cytokine storm induced by SARS-CoV-2 by using anti-IL-6 drugs is considered as a therapeutic option in several countries, with documented evidence for its efficiency [12, 13].

In Tunisia, medications that target the cytokine storm caused by COVID-19 (*tocilizumab and sarilumab*) are costly and not always available. In fact, they cannot be provided where indicated according to scientific evidence due to these factors. Determining an IL-6 cut-off score to predict fatal outcome could help physicians to decide who should benefit from these treatments, where available.

Significant correlations highlighted in our study are partially in accordance with Zhou J. and colleagues who reported similar correlations between serum IL-6 concentrations and age, urea, creatinine, NT-proBNP, cTroponin I, C-reactive protein and procalcitonin [14]. Zinc is an anti-inflammatory and antioxidant micronutrient available in food with a well-demonstrated relationship to immunity [15]. Zinc deficiency causes immunodeficiency with severe lymphopenia that is characterized by a considerable decrease in developing B cell compartments in the bone marrow [16]. A prospective study investigated the role of zinc deficiency in COVID-19 outcomes and found an OR of 5.54 for developing complications in zinc deficient COVID-19 patients (95% CI 1.56–19.6, p = 0.008) and 5.48 (95% CI 0.61–49.35, p = 0.129) for

**Table 4. Univariate analysis of survivors and non survivors.**

| | Non-Survivors (n = 43) | Survivors (n = 28) | p |
|---|---|---|---|
| Age, mean ± SD, years | 65.9 ± 7.4 | 60.5 ± 15.6 | 0.093 |
| Male gender, n (%) | 34 (70%) | 15 (30%) | 0.023 |
| SAPS II, med [IQR] | 32 [27–37] | 24 [21–33.7] | 0.004 |
| APACHE II, med [IQR] | 10 [7–14] | 8 [6–11.7] | 0.038 |
| IL-6, med [IQR], pg/ml | 70.8 [27.7–143] | 14.7 [3.4–33.3] | $< 10^{-3}$ |
| Zinc, med [IQR], µg/L | 51.5 [36.7–70.5] | 51.5 [32–69] | 0.0662 |
| WBC, med [IQR], $10^3/mm^3$ | 10.000 [7.100–13.600] | 8.650 [6.425–11.925] | 0.158 |
| Lymphocytes, mean ± SD, $10^3/mm^3$ | 1117 ± 517 | 1498 ± 1353 | 0.098 |
| Haemoglobin, mean ± SD, g/dL | 12.7 ± 2.3 | 12.5 ± 2.4 | 0.316 |
| Platelets, med [IQR], $10^3/mm^3$ | 266.000 [203.000–315.000] | 235.500 [193.750–288.000] | 0.129 |
| CPK, med [IQR], UI/L | 84 [43–183] | 171 [73–325] | 0.050 |
| LDH, med [IQR], UI/L | 450 [357–603] | 408 [308–613] | 0.488 |
| AST, med [IQR], UI/L | 26.5 [21–41.7] | 39.5 [27–48.7] | 0.048 |
| ALT, med [IQR], UI/L | 27.5 [19.7–40.5] | 27 [20–46] | 0.986 |
| CRP, med [IQR], mg/L | 115 [54–190] | 125 [56–187] | 0.725 |
| Troponin, med [IQR] | 0.01 [0–0.28] | 0.015 [0–0.08] | 0.332 |
| NT-proBNP, med [IQR], pg/ml | 36 [16.2–78] | 28 [11.4–51.2] | 0.105 |
| Fibrinogen, med [IQR], g/L | 6.36 [4.6–6.9] | 5.24 [4.6–6.9] | 0.974 |
| Lactates, med [IQR], meq/L | 1.6 [1–2] | 1.4 [1–1.9] | 0.776 |
| $PaO_2/FiO_2$, med [IQR], mmHg | 97 [83–152] | 134.5 [102–188] | 0.067 |
| DDimer, med [IQR], µg/L | 1040 [410–4340] | 630 [420–980] | 0.259 |
| Invasive MV, n (%) | 33 (76.7%) | 11 (39%) | 0.001 |

(SD: Standard Deviation; SAPS II: Simplified Acute Physiology Score II; APACHE II: Acute Physiologic and Chronic Health Evaluation II, CPK: creatine-phospho-kinase; LDH: lactico-deshydrogenase; AST: Aspartate aminotransferase; ALT: Alanine aminotransferase; IL-6: Interleukin-6; CRP: C-reactive protein; NT pro BNP: N-terminal pro-B type natriuretic peptide. MV: mechanical ventilation).

mortality [17]. We highlight in this study significant zinc serum deficiency among participants as well as a negative correlation with IL-6. Nevertheless, this correlation is statistically significant but fairly weak.

Since the severity of COVID-19 is related, in a large measure, to the extent of pulmonary involvement and consistent hypoxemia, the requirement for invasive mechanical ventilation was expected to be a fatal outcome predictor. Indeed, it simply means escalation after failure of non-invasive respiratory assistance among most severe patients.

Data on CPK, which is a marker of muscular damage, is only briefly mentioned in most papers on COVID-19. Muscle pain and fatigue are often reported during SARS-COV2

**Table 5. Multivariate logistic regression.**

| | p | OR | CI 95% |
|---|---|---|---|
| Male gender | 0.213 | 2.9 | 0.542–15.584 |
| SAPS II ≥ 28 | 0.160 | 4.02 | 0.578–27.989 |
| APACHE II ≥ 9 | 0.483 | 2.007 | 0.287–14.055 |
| IL-6 ≥ 20 pg/ml | $< 10^{-3}$ | 55.3 | 5.910–517.77 |
| CPK < 107 UI/L | 0.021 | 10.263 | 1.413–74.521 |
| AST < 30 UI/L | 0.022 | 9.37 | 1.372–64.069 |
| Invasive MV requirement | 0.042 | 6.444 | 1.066–38.972 |

infections, independent of severity. Elevated CPK may occur in COVID-19, but it remains unclear whether it is due to a virus-triggered inflammatory response or direct muscle toxicity [18]. Besides, in critically ill patients with acute respiratory failure, CPK elevation may result from additional respiratory muscle effort observable during respiratory distress. In a retrospective cohort study in Italy which included 331 COVID-19 patients, authors reported that increased CPK > 200 UI/L may predict a worse COVID-19 outcome [19]. Our data suggest that CPK < 107 UI/L is an independent marker of mortality risk. It appears that elevated CPK is a healthier host response against virus invasion. All studied patients had acute respiratory failure documented with hypoxemia in blood gases. Nevertheless, many of them were 'happy hypoxemic' and did not show any polypnoea or respiratory distress signs. Those patients often delay consulting, are tardily managed and had worse outcomes. So as a possible explanation, we have hypothesized that a lack of elevated CPK may be observed among these patients, but this remains to be confirmed.

## Strengths and limitations

Key strengths were prospective study design and quality correlations between IL-6 and other biological data measured in the same 24-hour period. To our knowledge, this is the first study providing IL-6 characteristics in COVID-19 critically ill patients managed in Tunisia and determining fatal cut-off, which may be helpful for determining treatment regimens and management in severe cases. Limitations are mainly represented by single-center design.

## Conclusions

The COVID-19 pandemic continues to threaten patients, societies, economies and healthcare systems around the world. During the first 24 hours of ICU admission, IL-6 ≥ 20 pg/mL as a cut off value is a predictor of fatal outcome. In low-income countries where IL-6 antagonists are not constantly available, dosing IL-6 at ICU admission may help physicians to decide who should mandatory benefit from these treatments. This can take part of a cost saving approach. In severe COVID-19 patients in ICU, we recommend serum level determination at admission to predict outcome and establish a therapeutic plan. Nevertheless, we need multicenter studies to improve our knowledge of the sensitivity and specificity of serum IL-6 cut-off values associated with fatal outcomes.

## Supporting information

**S1 Data.**
(SAV)

## Acknowledgments

We would like to thank all ICU team for its hard working during COVID-19 pandemic.

## Author Contributions

**Conceptualization:** Amira Jamoussi, Lynda Messaoud, Mohamed Besbes, Jalila Ben Khelil.

**Data curation:** Amira Jamoussi.

**Formal analysis:** Amira Jamoussi, Samia Ayed.

**Methodology:** Amira Jamoussi, Lynda Messaoud, Fatma Jarraya, Jalila Ben Khelil.

**Resources:** Emna Rachdi, Sadok Yaalaoui.

**Supervision:** Jalila Ben Khelil.

**Validation:** Mohamed Besbes, Jalila Ben Khelil.

**Writing – original draft:** Amira Jamoussi.

**Writing – review & editing:** Amira Jamoussi, Fatma Jarraya, Emna Rachdi, Nacef Ben Mrad, Sadok Yaalaoui, Mohamed Besbes, Samia Ayed, Jalila Ben Khelil.

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
