## [Decision Letter · Decision Letter 0]

11 Aug 2022

PONE-D-22-17377Interleukin6 prediction of mortality in critically ill COVID19 patients: a prospective observational cohort studyPLOS ONE

Dear Dr. JAMOUSSI,

Thank you for submitting your manuscript to PLOS ONE. After careful consideration, we feel that it has merit but does not fully meet PLOS ONE’s publication criteria as it currently stands. Therefore, we invite you to submit a revised version of the manuscript that addresses the points raised during the review process.

We look forward to receiving your revised manuscript.

Kind regards,

Yogesh Bahurupi, MD

Academic Editor

PLOS ONE

Journal Requirements:

2. In the ethics statement in the Methods, you have specified that verbal consent was obtained. Please provide additional details regarding how this consent was documented and witnessed, and state whether this was approved by the IRB.

4. We note you have included a table to which you do not refer in the text of your manuscript. Please ensure that you refer to Table 5 in your text; if accepted, production will need this reference to link the reader to the Table.

6.  We noticed you have some minor occurrence of overlapping text with the following previous publication(s), which needs to be addressed:

- https://pubmed.ncbi.nlm.nih.gov/32845568/

- https://link.springer.com/article/10.1007/s12011-020-02437-9?code=1df3a266-3ae1-4a34-8d0a-a94b5b77b46a&error=cookies_not_supported

In your revision ensure you cite all your sources (including your own works), and quote or rephrase any duplicated text outside the methods section. Further consideration is dependent on these concerns being addressed.

Additional Editor Comments:

I found this manuscript interesting.In introduction, the rationale and gap need to be more precisely addressed.

The discussion should bring out the implications and novelty of the study in a more robust way.

Reviewers' comments:

Reviewer's Responses to Questions

**Comments to the Author**

1. Is the manuscript technically sound, and do the data support the conclusions?

Reviewer #1: Yes

2. Has the statistical analysis been performed appropriately and rigorously? 

Reviewer #1: Yes

3. Have the authors made all data underlying the findings in their manuscript fully available?

Reviewer #1: Yes

4. Is the manuscript presented in an intelligible fashion and written in standard English?

Reviewer #1: Yes

5. Review Comments to the Author

Reviewer #1: In this manuscript, investigators examined clinical and laboratory parameters associated with COVID19 mortality among patients admitted to the ICU in Tunisia. They found that IL-6 measured within 24 hours to the ICU predicted subsequent ICU mortality. IL-6 correlated with other biomarkers including CRP, D-dimer and zinc. Invasive mechanical ventilation also predicted subsequent ICU mortality. In general, there are some minor issues but the study is technically sound. The results are not novel because similar findings have been reported in multiple other investigations.

1) Please specify the method used to determine the cutoff from the ROCs. It may be ideal to pick a value with higher sensitivity.

2) Please specify that the associations with IL-6 and other biomarkers is statistically significant but fairly weak. This possibly accounts for the lack of association among the other biomarkers and ICU mortality.

3) It is not specified whether the patients were mechanically ventilated within 24 hours or at any point during the hospitalization.

6. PLOS authors have the option to publish the peer review history of their article (what does this mean?). If published, this will include your full peer review and any attached files.

Reviewer #1: **Yes: **Manish Sagar

---

## [Author Response · Author response to Decision Letter 0]

13 Oct 2022

5. Review Comments to the Author

Reviewer #1: In this manuscript, investigators examined clinical and laboratory parameters associated with COVID19 mortality among patients admitted to the ICU in Tunisia. They found that IL-6 measured within 24 hours to the ICU predicted subsequent ICU mortality. IL-6 correlated with other biomarkers including CRP, D-dimer and zinc. Invasive mechanical ventilation also predicted subsequent ICU mortality. In general, there are some minor issues but the study is technically sound. The results are not novel because similar findings have been reported in multiple other investigations.

1) Please specify the method used to determine the cutoff from the ROCs. We selected the point on the ROC curve with the minimum distance from the left-upper corner of the unit square. It corresponds to IL-6 ≥ 47.7 pg/ml with sensitivity = 0.674 and specificity = 0.857

It may be ideal to pick a value with higher sensitivity. OK, we selected another point on the ROC curve with a distance slightly less than the first from the left-upper corner of the unit square. It corresponds to IL-6 ≥ 20 pg/ml with sensitivity = 0.837 and specificity = 0.679

This second cut-off has higher sensitivity but lower specificity.

We inserted this new value in the revised manuscript and modified the Multivariate logistic regression accordingly.

2) Please specify that the associations with IL-6 and other biomarkers is statistically significant but fairly weak. This possibly accounts for the lack of association among the other biomarkers and ICU mortality. OK, we specified in the discussion that the correlation was weak.

3) It is not specified whether the patients were mechanically ventilated within 24 hours or at any point during the hospitalization. It was already specified in ‘Management and outcome’:

Invasive mechanical ventilation was required in 44 (62 %) patients with a median initiation delay of 1 [0 - 4] days.

---

## [Decision Letter · Decision Letter 1]

19 Dec 2022

Interleukin6 prediction of mortality in critically ill COVID19 patients: a prospective observational cohort study

PONE-D-22-17377R1

Dear Dr. JAMOUSSI,

We’re pleased to inform you that your manuscript has been judged scientifically suitable for publication and will be formally accepted for publication once it meets all outstanding technical requirements.

Kind regards,

Jacopo Sabbatinelli, MD, PhD

Academic Editor

PLOS ONE

Additional Editor Comments (optional):

Reviewers' comments:

Reviewer's Responses to Questions

**Comments to the Author**

1. If the authors have adequately addressed your comments raised in a previous round of review and you feel that this manuscript is now acceptable for publication, you may indicate that here to bypass the “Comments to the Author” section, enter your conflict of interest statement in the “Confidential to Editor” section, and submit your "Accept" recommendation.

Reviewer #1: All comments have been addressed

2. Is the manuscript technically sound, and do the data support the conclusions?

Reviewer #1: Yes

3. Has the statistical analysis been performed appropriately and rigorously? 

Reviewer #1: I Don't Know

4. Have the authors made all data underlying the findings in their manuscript fully available?

Reviewer #1: Yes

5. Is the manuscript presented in an intelligible fashion and written in standard English?

Reviewer #1: Yes

6. Review Comments to the Author

Reviewer #1: Authors have addressed previous comments

I would have a statistics expert review the ROC analysis. In general there are strict criteria selected for selecting the cutoff.

7. PLOS authors have the option to publish the peer review history of their article (what does this mean?). If published, this will include your full peer review and any attached files.

Reviewer #1: **Yes: **Manish Sagar

---

## [Editor Report · Acceptance letter]

15 Feb 2023

PONE-D-22-17377R1 

Interleukin6 prediction of mortality in critically ill COVID19 patients: a prospective observational cohort study 

Dear Dr. JAMOUSSI:

I'm pleased to inform you that your manuscript has been deemed suitable for publication in PLOS ONE. Congratulations! Your manuscript is now with our production department. 

Kind regards, 

on behalf of

Dr. Jacopo Sabbatinelli 

Academic Editor

PLOS ONE